# An Integrated Genomic Strategy to Identify CHRNB4 as a Diagnostic/Prognostic Biomarker for Targeted Therapy in Head and Neck Cancer

**DOI:** 10.3390/cancers12051324

**Published:** 2020-05-22

**Authors:** Yi-Hsuan Chuang, Chia-Hwa Lee, Chun-Yu Lin, Chia-Lin Liu, Sing-Han Huang, Jung-Yu Lee, Yi-Yuan Chiu, Jih-Chin Lee, Jinn-Moon Yang

**Affiliations:** 1Institute of Bioinformatics and Systems Biology, National Chiao Tung University, Hsinchu 300, Taiwan; newyvmp40fm06@gmail.com (Y.-H.C.); chunyulin.bi99g@g2.nctu.edu.tw (C.-Y.L.); gb921.tw@gmail.com (S.-H.H.); leejungyu1012@gmail.com (J.-Y.L.); y2chiu@gmail.com (Y.-Y.C.); 2School of Medical Laboratory Science and Biotechnology, College of Medical Science and Technology, Taipei Medical University, Taipei 110, Taiwan; chlee@tmu.edu.tw; 3TMU Research Center of Cancer Translational Medicine, Taipei Medical University, Taipei 110, Taiwan; 4Ph.D. Program in Medical Biotechnology, College of Medical Science and Technology, Taipei Medical University, Taipei 110, Taiwan; 5Department of Biological Science and Technology, College of Biological Science and Technology, National Chiao Tung University, Hsinchu 300, Taiwan; 6Center for Intelligent Drug Systems and Smart Bio-devices, National Chiao Tung University, Hsinchu 300, Taiwan; 7Graduate Institute of Life Sciences, National Defense Medical Center, Taipei 110, Taiwan; im9527tw@yahoo.com.tw; 8Department of Otolaryngology-Head and Neck Surgery, Tri-Service General Hospital, National Defense Medical Center, Taipei 110, Taiwan; 9Faculty of Internal Medicine, College of Medicine, Kaohsiung Medical University, Kaohsiung City 807, Taiwan; 10Hepatobiliary Division, Department of Internal Medicine, Kaohsiung Medical University Hospital, Kaohsiung Medical University, Kaohsiung City 807, Taiwan

**Keywords:** head and neck squamous cell carcinoma (HNSCC), smoking, nicotine, prognostic biomarker, drug repurposing

## Abstract

Although many studies have shown the association between smoking and the increased incidence and adverse prognosis of head and neck squamous cell carcinoma (HNSCC), the mechanisms and pharmaceutical targets involved remain unclear. Here, we integrated gene expression signatures, genetic alterations, and survival analyses to identify prognostic indicators and therapeutic targets for smoking HNSCC patients, and we discovered that the FDA-approved drug varenicline inhibits the target for cancer cell migration/invasion. We first identified 18 smoking-related and prognostic genes for HNSCC by using RNA-Seq and clinical follow-up data. One of these genes, CHRNB4 (neuronal acetylcholine receptor subunit beta-4), increased the risk of death by approximately threefold in CHRNB4-high expression smokers compared to CHRNB4-low expression smokers (log rank, *p* = 0.00042; hazard ratio, 2.82; 95% CI, 1.55–5.14), former smokers, and non-smokers. Furthermore, we examined the functional enrichment of co-regulated genes of CHRNB4 and its 246 frequently occurring copy number alterations (CNAs). We found that these genes were involved in promoting angiogenesis, resisting cell death, and sustaining proliferation, and contributed to much worse outcomes for CHRNB4-high patients. Finally, we performed CHRNB4 gene editing and drug inhibition assays, and the results validate these observations. In summary, our study suggests that CHRNB4 is a prognostic indicator for smoking HNSCC patients and provides a potential new therapeutic drug to prevent recurrence or distant metastasis.

## 1. Introduction

Smoking, alcohol consumption, and human papillomavirus (HPV) infection are considered to be the major risk factors for increasing the incidence and mortality rates of head and neck squamous cell carcinoma (HNSCC) [1,2]. Currently, there are more than 600,000 new cases of HNSCC yearly, and the mortality rate is 40–50% worldwide [1]. HPV-positive (HPV+ve) HNSCCs are considered far more favourable than HPV-negative (HPV-ve) HNSCCs in terms of prognosis, and HPV+ve cases account for approximately 20% of all HNSCC cases [3,4,5,6]. The most common risk factor is tobacco smoking; at least 70–80% of cases of HNSCCs are attributable to smoking alone or a combination smoking with additional risk factors (e.g., alcohol or areca nut chewing) [5,7].

The role of smoking for HNSCC has been studied in biochemistry, genetic alteration, and therapeutic response. For example, the carcinogens and their metabolites (e.g., nicotine-derived nitrosamine ketone (NNK)) of tobacco smoking can generate formation of DNA adducts by binding covalently to DNA, causing genetic mutation and resulting in HNSCC [8,9]. In addition, having a history of tobacco smoking is an established risk factor for determining the incidence, survival, and therapeutic response for HNSCC [10]. HNSCC incidence is approximately 10- to 20-fold higher in heavy smokers (smoking alone) than in patients who have never smoked [11,12,13]. Furthermore, there is no safe dose for tobacco use, and even a low smoking frequency (>0–3 cigarettes per day) increases the HNSCC incidence by 50% [14]. Tobacco smoking is considered a strong prognostic indicator [7,15,16], which significantly increases the risk of death ≈2-fold for smokers vs. non-smokers [7,17,18,19,20] and ≈1.5-fold for HPV+ve smokers vs. HPV+ve non-smokers [21,22].

Nicotine is one of the major components of tobacco, and it primarily acts on nicotinic acetylcholine receptors (nAChRs) and causes dependence in the brain [23,24]. Nicotine addiction induced by nAChRs is a major difficulty in smoking cessation. Therefore, to decrease the pleasurable effect from the brain’s reward system, current smoking cessation agents (e.g., varenicline, bupropion, and mecamylamine) try to compete with nicotine to bind to the receptors [25,26,27,28,29]. Furthermore, some clinical studies found that HNSCC patients who ceased smoking after diagnosis or before chemoradiation were associated with better therapeutic response and prognosis [30,31,32]. One of the studies reported that patients with higher nAChR subunits CHRNA5 transcript levels were associated with lower effectiveness of radiotherapy and worse prognosis. Previous HNSCC studies also showed that nicotine and its metabolites promoted tumour cell growth and migration. It has been found that nAChR subunits displayed different protein expression in different tumour sites promoting tumour progression, such as CHRNA1, which increased protein expression in the hypopharynx, larynx, and advanced tumours, but decreased in oropharynx tumours [33]. Another study indicated that CHRNA7 could activate EGFR and NF-κB downstream signalling, such as PI3K-AKT and Snail-RKIP pathways that promote HNSCC cell line growth and migration [34,35]. However, some studies were against the conclusion that the activation of PI3K-AKT is not dependent upon CHRNA7, which was probably due to the discrepancy of sample type (clinical tissue vs. cell lines), different cell lines, or concentration of nicotine or agent [36,37]. Although the risk of smoking in HNSCC had be established, the molecular mechanism of nicotine induced in tumour progression of HNSCC remains obscure. In addition, nicotine-derived nitrosamine ketone (NNK) is also a major compound of tobacco smoke, showing higher binding affinity than nicotine for nAChRs in lung cancer [38,39,40,41]. Currently, NNK is considered a potent carcinogen for multiple cancers, such as HNSCC, lung cancer, oesophageal cancer, breast cancer, and bladder cancers [34,42,43,44,45,46,47].

Because of the advantages of investigating genomic variation, heterogeneity, and clinical outcomes from omics data [5,48,49,50,51,52], studies of many cancers including HNSCC have shifted the focus to discover signatures for the prediction of disease incidence, diagnosis, subtype, prognosis, and treatment response with the goal of personalized treatment. However, HNSCC is an unexpectedly heterogeneous cancer due to the multiple risk factors, complex anatomy of tumour sites, and large genetic alterations that drive tumour progression. Therefore, HNSCC needs more promising biomarkers to accurately identify characteristics of subtypes and clinical outcomes.

Due to large amount of time and high costs involved in finding new therapeutic drugs, drug repurposing strategies are increasingly utilized to provide faster and safer clinical trials, as well as to reduce failure and cost [53,54,55]. Many studies show that several drugs that are already approved offer novel therapeutic options for cancers alone or in combination with other agents. Successfully repurposed drugs include thalidomide for myeloma, aspirin for colorectal cancer, and metformin for liver or breast cancer [56]. Currently, the major treatment approaches for HNSCC remains surgery, radiotherapy, chemotherapy, and EGFR-specific antibodies (e.g., chemoradiotherapy or concomitant with cetuximab) [57,58]. EGFR-specific antibody combines with radiotherapy or chemotherapy that controls locoregional recurrences and reduces mortality [57,59]. However, about 60% studies showed the associated between EGFR overexpression and adverse prognosis, whereas 40% did not [4]. Therefore, HNSCC needs new biomarkers and corresponding therapy agents to provide more choices besides the current options.

In this study, we aimed to identify biomarkers for smoking HNSCC patients and potential agents. Primarily, we systematically integrated the genomic and clinical data of HNSCC patients (Figure 1) and found neuronal acetylcholine receptor subunit beta-4 (CHRNB4), a kind of nAChR, whose gene expression was upregulated and associated with adverse prognosis in smoking patients. Furthermore, on the basis of our previous work and drug repurposing approach, “Homopharma” [60], we discovered that an FDA-approved smoking cessation agent, varenicline, inhibits CHRNB4 in long-term NNK-treated HNSCC cells. Our results indicate that CHRNB4 is able to determine whether patients who smoke have a favourable or adverse prognosis and that the invasion and migration of NNK-treated cells were decreased when CHRNB4 was inhibited.

## 2. Results

### 2.1. Identification of Smoking-Related and Prognostic Genes in Head and Neck Cancer

To identify smoking-related and prognostic biomarkers, patients were stratified into 175 smokers, 117 non-smokers, and 120 low-survival patients with smoking history and clinical follow-up from The Cancer Genome Atlas (TCGA) database (Figure 2A). By comparing the patient groups and corresponding normal samples, we identified 480 differentially expressed genes (DEGs), with fold change ≥1.5 and *p*-value < 0.05, which were considered candidate genes related to smoking-induced HNSCC and poor clinical outcomes.

To analyse the association between these 480 selected genes and 5-year overall survival (OS), we utilized the median value (50%) of each gene’s expression to classify patients into two distinct high- and low-risk groups by Kaplan–Meier analysis (Figure 2B). The results showed that 54 genes were able to detect significant survival differences in these two groups using the log-rank test (*p* < 0.05, red and blue dots). Furthermore, Cox proportional hazards regression analysis was also used to obtain hazard ratios (HRs) with 95% confidence intervals for determining favourable (HR < 1, blue dot) and adverse prognostic genes (HR > 1, red dot). Here, we focused on 18 adverse prognostic genes, such as DKK1, CHRNB4, TRIML2, IFIT1, and BASP1, as potential therapeutic targets and diagnostic biomarkers (Appendix A). Among these 18 genes, only four genes (i.e., CHRNB4, GPC6, ORAOV1, and PPFIA1) were differentially expressed between smokers vs. normal samples but were not differentially expressed between non-smokers vs. normal samples (Figure 2C). On the basis of our previous work and domain knowledge, we selected CHRNB4 for further analysis, and Western blotting was performed to validate CHRNB4 expression in NNK-treated HNSCC cells.

CHRNB4 gene expression is not only specific for smoking patients but also predicts prognosis when patients are classified into four groups (Figure 2D). CHRNB4 expression showed the most significant difference in the overall survival (OS) for CHRNB4 high-expression smokers (red) compared with the other three groups, which were CHRNB4 low-expression smokers (orange), former smokers (blue), and non-smokers (black). Kaplan–Meier analysis and log-rank test showed that the OS probability of the CHRNB4-high subgroup (red) was the lowest among these four subgroups, especially compared to CHRNB4-low (*p* = 4.2 × 10^−4^, HR = 2.82). However, no statistically significant differences were observed in OS probability when comparing CHRNB4-low to other subgroups. For additional verification of the association of CHRNB4 gene expression and prognosis and smoking behaviour, boxplots were generated. The boxplots of 87 high-survival (favourable outcome) and 120 low-survival patients (adverse outcome; see Materials and Methods section) showed that the CHRNB4 gene expression of smokers was significantly higher than that of non-smokers in the low-survival group (Student’s *t*-test, *p* = 0.001), but there were no differences in the high-survival group (Appendix A). Upon further investigation, we used immunohistochemistry (IHC) stain to reveal the CHRNB4 expression on clinical tissues from smoking or non-smoking HNSCC patients (Figure 2E). With duplicate patient IHC analysis, it is easy to identify that CHRNB4 is intensively expressed in the membrane of cancerous region from smoking HNSCC patients (black arrowed) compared to non-smoking HNSCC patients. In addition to this, the normal region in adjacent tumour tissue also showed very low CHRNB4 expression (green arrowed). These results support the idea that CHRNB4 is a potential biomarker associated with smoking and prognosis in HNSCC.

### 2.2. Association Cancer-Related Genes and CHRNB4

To understand the association of tumorigenesis and poor prognosis with high CHRNB4 expression, we first collected 93 pathways with 2652 genes involved in cancer hallmarks from the Atlas of Cancer Signalling Network (ACSN) database [61] and 3530 cancer-related genes from the DisGeNET database (Appendix A) [62]. Next, we calculated the Pearson correlation coefficient to identify potential co-regulated genes of CHRNB4 on the basis of RNA-Seq data of the 87 CHRNB4-high and 88 CHRNB4-low smoking patients. We then considered 68 (CHRNB4-high, red bars) and 23 (CHRNB4-low, blue bars) co-expressed genes with |Pearson’s *r*| ≥ 0.4 and investigated their enriched pathways (Figure 3A and Appendix A). Among these 91 genes, 12 (CHRNB4-high, red) and 1 (CHRNB4-low, blue) genes are related to cancer hallmarks (from ACSN), and 18 (CHRNB4-high) and 4 (CHRNB4-low) are cancer-related genes (from DisGeNET). The number of cancer-related genes co-expressed with CHRNB4 in the CHRNB4-high group was found to be more than that of the CHRNB4-low group, with odds ratios of 4.71 for ACSN and 1.71 for DisGeNET (Figure 3A). We then performed enrichment analysis by using a hypergeometric distribution to investigate enriched pathways that are related to cancer hallmarks and recorded in the Kyoto Encyclopedia of Genes and Genomes (KEGG) (Figure 3B,C and Equation (1)). In the cancer hallmark dataset, the number of co-expressed genes in the CHRNB4-high subgroup was equal to or greater than that in the CHRNB4-low subgroup (Figure 3B), and 11 enriched KEGG pathways (hypergeometric test *p* < 0.05 and ≥3 genes involved in pathways/modules) were identified for the CHRNB4-high subgroup but none were identified for the CHRNB4-low subgroup. For example, according to the enrichment results and KEGG “pathway in cancer” (hsa05200), ADCY9 regulates cGMP-PKG and cAMP signalling pathways to inhibit apoptosis and promote proliferation [63]; the genes NOCA1 and NOCA3 are involved in oestrogen signalling, breast cancer pathway, and cancer cell growth stimulation [64,65]; the genes ARNT and NOTCH3 enhance migration and invasion via angiogenesis by way of endocrine resistance, HIF-1, and notch signalling [66,67] (Figure 3C). The results indicate that CHRNB4-high patients are often to promote proliferation and migration and inhibit apoptosis, leading to poor prognosis.

### 2.3. Relationship between Genetic Alterations and CHRNB4 Signature

To investigate the relationship between genetic alterations and the CHRNB4 signature for smoking HNSCC patients, we identified 246 genes with significantly different copy number alterations (CNAs) between 87 CHRNB4-high and 88 CHRNB4-low smoking patients by using Fisher’s exact test (only 83 of 87 CHRNB4-high patients’ copy number data were obtained from the database, details are found in the Materials and Methods section and Equation (2)). The results show that the alteration frequencies of 246 genes were largely higher in CHRNB4-high patients than those in the other two subgroups, CHRNB4-low (Student’s *t*-test, *p* = 1.71 × 10^−105^) and non-smoker (Student’s *t*-test, *p* = 5.92 × 10^−95^) groups (Figure 4A). In addition, 242 of 246 genes displayed approximately 6–25% (median: 20%) more frequent alterations in the CHRNB4-high expression subgroup than the frequency in the CHRNB4-low expression subgroup (Figure 4B and Appendix A). This result indicates that the genome was more instable in the smokers with high CHRNB4 expression than that of the smokers with low CHRNB4 expression. For example, in TPRG1 associated with HNSCC potential oncogene TP63, LPP played the role of cancer cell migration, and CCDC50 was essential for cancer cell survival. Interestingly, the frequency of amplification was always greater than or equal to the frequency of deletion for these 246 CNAs in both CHRNB4-high and CHRNB4-low smokers, that is, the distributions of the copy number deletion between two groups were not significantly different for 246 genes.

Of these 246 genes, 34 were recognized in the eight hallmarks of cancer (Figure 4C), displaying similar results as the genes that were co-expressed with CHRNB4, which sustained cancer cell survival and invasion/migration pathways (Figure 2B). In addition, we identified seven significantly enriched KEGG pathways (hypergeometric test *p* < 0.05 and ≥3 genes involved in pathway/module; Figure 4D). For example, the pathway of the disease of type II diabetes mellitus, is correlated to tobacco smoking, and the genes involved are ADIPOQ, PIK3CA, and SLC2A2 [68,69]. The serotonergic synapse pathway was observed in smoking-related cancer, and the serotonin was found to promote tumour growth in liver cancer and stimulated cell cycle progression in HNSCC [70,71]. The pathways of choline metabolism in cancer and leukocyte transendothelial migration are related to cell proliferation, migration, angiogenesis, and inflammation in tumorigenesis. These results provide the hints of the associations of adverse prognosis and smokers with high CHRNB4 expression due to activating more multiple hallmark functions of cancer.

### 2.4. The Pathways and Downstream Targets of CHRNB4-High and -Low Subgroups

We reconstructed a simplified network as the reference template to illustrate signalling interactions among the common and different pathways regulated by smoking patients with CHRNB4-high expression and CHRNB4-low expression (Figure 5) according to “pathways in cancer” (KEGG: hsa05200). The downstream targets, such as HES1 and IL12A (green border) inducing angiogenic signalling (left side), displayed genetic alterations predominantly in the CHRNB4-high subgroup. Additionally, the co-expressed genes (e.g., ARNT and NOTCH) of CHRNB4 in the CHRNB4-high group (pink) and the gene FLT4 in the CHRNB4-low subgroup (orange) were involved in HIF-1 and Notch signalling pathways, which promote angiogenesis and migration in smoking HNSCC patients.

Moreover, KNG1 and GNB4 (green border), with their downstream targets PLD1 and PIK3CA (middle diagram), enhanced cell proliferation and evading apoptosis in CHRNB4-high patients through the Ras and PI3K-Akt signalling pathways. On the other hand, the telomere length-associated gene TERC and oxidative stress response gene NFE2L2 were more frequently amplified in the CHRNB4-high subgroup, which could immortalize cells and against oxidative stress for promoting cancer cell survival and escaping apoptosis. The co-expressed gene ADCY9 in the CHRNB4-high group and the HNSCC oncogene FADD, which had alterations in both subgroups, also contributed to the inhibition of apoptosis. Additionally, to sustain proliferative signalling (right side), the tumour oncogene CCND1, probably influenced by the co-expressed gene NCOA1/3 as well as the suppressors CDKN2A and CDKN2B, which govern the cell cycle, were genetically altered to enhance cancer cell growth in both smoking subgroups. Additionally, FGF12, DVL3, and MECOM (green border) were more often altered in CHRNB4-high patients for sustaining cell proliferation.

On the basis of the analysis and results above, we summarized several key observations: (1) there were 68 genes co-expressed with CHRNB4 in the CHRNB4-high subgroup and 23 in the CHRNB4-low subgroup; (2) there were 246 significantly different CNAs between the CHRNB4-high and -low subgroups; (3) common amplifications and deletions (e.g., FGFs, MYC, CCND1, CDKN2A, and CDKN2B) occurred in both CHRNB4-high and -low patients. These results show that CHRNB4 plays a key role in regulating cancer-related functions and affecting the clinical outcome of smoking HNSCC.

### 2.5. Validation of CHRNB4 Gene Editing in HNSCC Cells

To validate the impact of NNK treatment on CHRNB4, we generated long-term NNK-treated FaDu, SCC25, and OECM1 HNSCC cell lines (Figure 6A). After 1 month of NNK treatment, both FaDu and SCC25 cell lines had significantly higher CHRNB4 protein expressions than the parental controls (without NNK treatment), whereas long-term NNK-treated OECM1 obtained a minor CHRNB4 protein induction compared with the parental control. To understand the tumorigenic role of CHRNB4 during HNSCC development, we performed CRISPR/Cas9 gene editing for CHRNB4 with two independent sgRNAs on both long-term NNK-treated FaDu and SCC25 cells (Figure 6B,C). The CHRNB4 protein expressions of both cell lines were significantly reduced in CHRNB4 sgRNA-1 and sgRNA-2 virus-transfected cells, compared to controls and scramble-sgRNA controls (SC). Then, the CHRNB4 gene loci were Sanger-sequenced, showing clear and single DNA signals on both scramble-sgRNA transfected FaDu (Figure 6D) and SCC25 (Figure 6E) cells. On the other hand, with CHRNB4 sgRNA-2 virus transfection, the CHRNB4 gene sequences were significantly edited, showing the great gene editing efficiency on both FaDu (Figure 6F) and SCC25 (Figure 6G) cells (sgRNA-2 used for further analysis, due to the high gene editing efficiency). Furthermore, to analyse CRISPR-based edit efficiency, Tracking of Indels by Decomposition (TIDE) was used to reveal the insertions and deletions (indels) that induced in FaDu and SCC25 cells with CHRNB4 sgRNA-2 introduction [72]. The result showed the average CHRNB4 gene editing efficiency on FaDu cells was 88.6% (Figure 6H,I), containing 58.4% and 21.8% with single-base insertion and other editing patterns, respectively. On the other hand, CHRNB4 gene editing efficiency on SCC25 cells was 94% (Figure 6J,K), containing 49.5% and 38.9% with single-base insertion and other editing patterns, respectively.

### 2.6. CHRNB4 Promoted Cell Migration and Invasion in NNK-Treated HNSCC Cells

Next, we used cell migration and invasion assays to investigate the tumorigenic role of CHRNB4 on long-term NNK-treated FaDu and SCC25 cells (Figure 7 and Appendix A), compared to cells without NNK treatment. Both FaDu (Figure 7A,C) and SCC25 (Figure 7B,D) cell lines displayed significantly higher migration ability of 1.45-fold and 1.58-fold when they were treated with 0.1 μM NNK, respectively. On the other hand, both FaDu (Figure 7E,G) and SCC25 (Figure 7F,H) cell lines displayed significantly higher invasion ability of 5.53-fold and 2.71-fold in NNK-treated cells, respectively. However, when CHRNB4 was gene edited on FaDu and SCC25 cells, both migration and invasion abilities were significantly reduced in both treatment groups with NNK and treatment groups without NNK. These results indicate that CHRNB4 not only plays important roles in NNK-induced tumorigenesis in HNSCC cells, but also implies CHRNB4 expression is essential during HNSCC development. Therefore, CHRNB4 could be a potential druggable target to prevent metastasis in HNSCC, especially for smoking HNSCC patients.

### 2.7. Varenicline, as a Repurposing Drug of CHRNB4, Inhibited Migration and Invasion in HNSCC Cell Lines

On the basis of the results above, we considered the idea that CHRNB4 could be a potential therapeutic target for the inhibition of migration and invasion in smoking HNSCC patients. By using in-house methods and tools (Homopharma and iGEMDOCK) [60,73,74], we discovered an FDA-approved drug, called varenicline, as a potential CHRNB4 inhibitor (purchased from Sigma-Aldrich, #PZ0004). We utilized parental cell lines (FaDu and SCC25) and NNK-treated cell lines to examine the efficacy of the inhibition of migration and invasion by treating cells with varenicline at concentrations of 0, 0.1, 1, and 5 μM (Figure 8 and Appendix A). The results showed that NNK significantly induced migration and invasion abilities on both parental FaDu (Figure 8A,E) and SCC25 (Figure 8C,G) cells, whereas varenicline significantly inhibited FaDu and SCC25 cell migrations and invasions at 1 and 5 μM during varenicline treatment. Consistently, varenicline significantly decreased migration and invasion abilities in NNK-treated FaDu and SCC25 cell lines with 1 and 5 μM during varenicline treatment, especially in the SCC25 cell line. The NNK-treated cells with varenicline treatment were significantly decreased for 38% migration and 61% invasion abilities in FaDu (Figure 8B,F), whereas they reduced 42% migration and 84% invasion abilities in SCC25 cell lines (Figure 8D,H) at 5 μM concentration. This result indicates that varenicline has great pharmacological potential to inhibit metastasis ability in both FaDu and SCC25 cells, and that varenicline demonstrates even greater inhibition efficacy in NNK-treated HNSCC cells. Therefore, the use of varenicline may act as a novel anti-metastasis agent for smoking HNSCC patients in the clinic. In the present study, integration of the analysis of gene expression, overall survival, cancer-related pathways, and the experiments of CHRNB4 gene editing and drug treatment suggested that CHRNB4 is a promising prognostic biomarker and drug target for smoking HNSCC patients.

## 3. Discussion

Here, we identified the CHRNB4 signature as a prognostic predictor in smoking HNSCC patients. Patient smoking status was significantly associated with adverse prognosis in the CHRNB4-high subgroup compared to the CHRNB4-low subgroup. In the CHRNB4-high subgroup, CHRNB4 was co-expressed with numerous genes, such as ADCY9, NOTCH3, ARNT, NCOA1, and NCOA3, which correlated with multiple cancer-related processes [63,64,65,66,67], but only one gene, FLT4, was co-expressed in the CHRNB4-low subgroup [75]. These results suggest that CHRNB4-high patients who smoke have poor clinical outcomes due to this gene association with more genes involved in cancer-related processes.

According to the KEGG “pathways in cancer” (hsa05231), we integrated co-expression and CNA analysis to find dysregulated signalling pathways and downstream targets. In both CHRNB4-high and -low subgroups, we observed deep copy number deletion of the tumour-suppressor genes CDKN2A and CDKN2B in ≈40% of smoking patients [49] and high-level copy number amplification of the oncogenes FADD and CCND1 in ≈35% of smoking patients [49], which contributed to cancer cell survival and proliferation in smoking patients. Furthermore, there were 11 genes with alterations that were more frequent in the CHRNB4-high subgroup than in the CHRNB4-low group. On the basis of the hallmarks of cancer and KEGG pathways, these 11 genes are primarily involved in angiogenesis, resisting cell death, and sustaining proliferation; for instance, HES1 and IL12A perturb the Notch and the Jak-STAT signalling pathways, respectively, which are involved in angiogenesis in the CHRNB4-high subgroup. Other alterations, such as KNG1, PLD1, GNB4, PIK3CA, TERC, NFE2L2, DVL3, FGF12, and MECOM, which are also dysregulated in other cancers [76,77,78,79,80,81,82,83,84,85], showed co-amplification in CHRNB4-high patients, promoting cell survival and growth. Interestingly, previous studies showed that the oncogene PIK3CA was altered in both HPV-ve and HPV+ve patients (34% and 56%, respectively, predominantly in HPV+ve) and contributed to evading apoptosis [4,49,50]. However, in our CNA results, PIK3CA had more frequent genetic alterations in CHRNB4-high patients than in CHRNB4-low patients (30% and 9%, respectively, Fisher’s exact test *p* < 0.05). Considering that the major aetiology of HPV-ve HNSCC is smoking, we propose that CHRNB4 is able to identify new HPV-ve subtypes in smoking patients. Due to the missing HPV status values in TCGA clinical data, the true HPV status of most patients is not available, and this assumption remains to be fully confirmed. In summary, we suggest that the CHRNB4 signature identified a new smoking subgroup with a much worse prognosis.

In our previous work [86], we identified nicotinic acetylcholine receptors (nAChRs) and their corresponding inhibitors from FDA-approved drugs (e.g., CHRNB4 inhibitors, varenicline; CHRNA9 inhibitors, bupropion) by using our Homopharma approach and the *i*GEMDOCK tool (data not shown) [60]. The drug inhibition assay showed that varenicline could significantly control cell migration and invasion in long-term NNK-treated cells in a dose-dependent manner, but it had limited effects on parental cell lines. On the basis of the results of the CHRNB4 gene editing and drug inhibition assay, we considered varenicline to be a potential targeted drug that may reduce cancer metastasis in smoking HNSCC patients who have high CHRNB4 expression. However, the detailed mechanisms by which nicotine and varenicline compete with nAChRs (e.g., CHRNB4) to regulate downstream signalling pathways and metastasis in smoking patients and in NNK-treated cells remain to be elucidated. We believe that CHRNB4 and the repurposed drug varenicline will be useful for the development of precision medicine.

Our study has several limitations. First, we did not acquire the HPV status of each sample from the TCGA clinical data. There were 87 patients in the CHRNB4-high subgroup (88 patients in the CHRNB4-low subgroup), but only 16 HPV-ve and 6 HPV+ve patients (15 HPV-ve and 5 HPV+ve in the CHRNB4-low subgroup) were identified by using p16 testing; the HPV status of other patients was not available in the database (Appendix A). Although smoking is a major aetiology for HPV-ve HNSCC, there were no significant differences in HPV status between the CHRNB4-high and -low subgroups. Therefore, the role of CHRNB4 in HPV+ve or HPV-ve patients remains to be further explored. Second, we relied on smoking history provided by the TCGA to identify current smokers, lifelong non-smokers, and former smokers, and these data were acquired at diagnosis. However, other studies may stratify smokers by different rules that may produce slight differences in our results. In addition, on the basis of the TCGA clinical data, it is difficult to judge whether former smokers actually avoid the harms of tobacco smoke or second-hand smoke from people around them without biochemical confirmation. Third, although varenicline is primarily used for smoking cessation, varenicline has low efficacy in the inhibition of SCC25 and FaDu cell viability when cells are treated with or without NNK, as detected by using the MTT assay (data not shown), and varenicline only inhibited cell migration and invasion in NNK-treated cells.

## 4. Materials and Methods

### 4.1. Patient Stratification and RNA-Seq Data Collection

In total, we collected level 3 RNA-Seq datasets with 522 HNSCC tumour samples and 44 corresponding normal tissues from The Cancer Genome Atlas (TCGA). Next, 504 patients, whose data included gene expression data of primary solid tumours (tumour type: 01) and sufficient clinical data (e.g., follow-up time and tobacco smoking history), were considered in this study (Appendix A). Tobacco smoking history was used to annotate patients who were smokers, former smokers, and non-smokers by referencing the NIH Common Data Element (CDE) Resource Portal. Ultimately, 175 current smokers, 117 lifelong non-smokers, and 212 former smokers were annotated for identifying prognostic and smoking-related genes.

Furthermore, patients were divided into “low-survival” (adverse prognosis) and “high-survival” (favourable prognosis) groups to study the relationship between smoking and clinical the outcome of HNSCC. The “high-survival” group was defined by the TCGA HNSCC study and censored alive for at least the 75th percentile of total follow-up time. The “low-survival” group was composed of patients who died within the a time frame that was lower than the 75th percentile of total follow-up duration in the HNSCC study [87]. For example, the total duration of follow-up of the 504 HNSCC patients was 17.58 years, and thus the 75th percentile observation time corresponded to 2.37 years. The patients who died within 2.37 years were considered the “low-survival” group, whereas the “high-survival” group was composed of the patients who were censored alive for at least 2.37 years of follow-up time. Finally, 120 low-survival and 87 high-survival patients were used to measure the gene expression differences that were related to smoking status.

### 4.2. Methods Overview

To identify HNSCC smoking-related biomarkers and potential targeted agents, we first identified 480 differentially expressed genes (DEGs) by comparing smokers vs. normal, low-survival patients vs. normal, and smokers vs. non-smokers. We then used the Cox proportional hazards regression to identify 18 adverse genes and 36 favourable genes in smoking HNSCC patients. Among these genes, we selected CHRNB4, a nAChR that is strongly associated with smoking, and we utilized its gene expression to classify smoking patients to reveal the reasons for adverse prognosis in smokers with high CHRNB4 expression. Therefore, we computed the co-expressed genes of CHRNB4 and the differential copy number alterations (CNAs) between CHRNB4-high and CHRNB4-low subgroups. Cancer hallmark analysis was also performed on these genes to explore the relationship between dysregulated pathways and clinical outcome. Finally, migration, invasion, gene editing, and drug inhibition assays were performed to verify whether CHRNB4 is a potential HNSCC smoking-related prognostic biomarker (Figure 1).

### 4.3. Identification of DEGs

We used a log2-transformation to compute the TCGA level 3 RNA-Seq data and generate gene expression profiles of 20,531 genes for 504 samples. In this analysis, the fold change and a modified *t*-statistic (limma package v. 3.38.3) were utilized to identify DEGs between corresponding normal tissues and tumours in patients with different smoking statuses and prognoses. Finally, the genes with |fold change| ≥ 1.5 and *p*-value < 0.05 were considered as DEGs and smoking-related and prognostic biomarker candidates.

### 4.4. Survival Analysis

We used RNA-Seq data with complete survival and smoking status to acquire the association between prognostic significance and CHRNB4 expression. Overall survival (OS) was defined as the time from the date of diagnosis to death and was censored at the last follow-up for alive patients. Here, we utilized the median value of the RNA-Seq expression to divide smoking patients into high- and low-expression groups to perform Kaplan–Meier analysis of their association with 5 year survival. Moreover, we used the log-rank test and Cox proportional hazards regression analysis of each gene to determine adverse or favourable prognostic significance. For example, the median RNA-Seq value of CHRNB4 in 175 smokers was 3.94, then smokers whose CHRNB4 expression was higher than 3.94 were grouped into CHRNB4-high subgroup, otherwise they were grouped into the CHRNB4-low subgroup. Next, the univariate Cox proportional hazards regression was performed for two smoking subgroups with high and low gene expression. (Figure 2B, and the results of multivariate Cox proportional hazards regression are shown in Appendix A). The Kaplan–Meier analysis, log-rank test, and Cox proportional hazards regression analysis were performed by using the R survival (v. 2.37.2) and survplot (v. 0.0.7) packages.

### 4.5. Cancer Hallmark and Pathway Enrichment Analysis

To study how CHRNB4 affects tumorigenesis and prognosis in smoking HNSCC patients, we first collected 93 biological processes including 2652 genes involved in 10 hallmarks of cancer, as well as 330 human pathways including 7469 genes with corresponding interactions from the Atlas of Cancer Signalling Network database (ACSN, https://acsn.curie.fr/ACSN2/downloads.html) and Kyoto Encyclopedia of Genes and Genomes (KEGG) [78,88], respectively. In addition, we also determined 3530 cancer-related genes by using keywords, including “cancer”, “carcinoma”, and “malignant tumor/neoplasm/neoplasms” from the DisGeNET database for further analysis (Appendix A) [62]. We utilized gene ID numbers (National Center for Biotechnology Information) to integrate and map genome annotations across different databases. Next, we used Pearson’s correlation coefficient to compute the genes co-expressed with CHRNB4 in CHRNB4-high and CHRNB4-low expression patients who smoke. Finally, 68 and 23 genes were selected by |Pearson’s *r*| ≥ 0.4 from two subgroups to further estimate the relationships between CHRNB4 and tumorigenesis and prognosis by using enrichment analysis. Furthermore, the genes whose copy numbers were altered between CHRNB4-high and CHRNB4-low smoking patients underwent functional enrichment analysis to clarify the processes and pathways associated with CHRNB4 in smoking-related HNSCC. The enrichment *p*-value was measured by hypergeometric distribution on the basis of the probabilities of genes involved in certain modules or pathways and considered more likely to provide reasons for enhancing tumour progression and adverse prognosis. The enrichment *p*-value of the hypergeometric distribution is defined as
(1)$$P=\overset{n}{\underset{i=x}{\sum }}\frac{\left(\begin{array}{c} M\\ i \end{array}\right)\left(\begin{array}{c} N-M\\ n-i \end{array}\right)}{\left(\begin{array}{c} N\\ n \end{array}\right)}$$
where *i* is the number of the genes co-expressed with CHRNB4 involved in certain gene sets or pathways, and *n* is the total number of the co-expressed genes of CHRNB4 with |Pearson’s *r*| ≥ 0.4 in a specific subgroup; *x* represents the genes whose co-expression has |Pearson’s *r*| ≥ 0.4, for example, *x* and *n* are separately 8 and 68 genes that were recorded in Sustaining Proliferative Signalling gene sets and identified in CHRNB4-high patients who smoke (Figure 3C); *M* and *N* are the numbers of the total members in certain gene sets/pathways and total genes, respectively, used for the comparison of tumorigenesis between CHRNB4-high and CHRNB4-low expression patients who smoke.

### 4.6. Copy Number Alteration (CNA) Analysis

To investigate the association between CHRNB4 and genetic alteration in smoking patients, we obtained the 20,316 DNA copy number data of TCGA from the cBioPortal database (https://www.cbioportal.org/datasets) [89,90], which generates their calls by the GISTIC 2.0 algorithm [91,92]. Here, the 16,915 genes altered in more than one sample with deep deletion or high-level amplification (gene annotation is “−2” and “2”, respectively) were considered CNA genes that are biologically relevant for HNSCC. For each gene, we estimated the distribution of CNA between CHRNB4-high and CHRNB4-low patients who smoke by using Fisher’s exact test, defined as
(2)$$P=\frac{\left(\begin{array}{c} a+b\\ a \end{array}\right)\left(\begin{array}{c} c+d\\ c \end{array}\right)}{\left(\begin{array}{c} n\\ a+c \end{array}\right)}$$
where *a* and *b* are the numbers of the patients with a gene copy number alteration (i.e., gene annotation is “−2” or “2”) in CHRNB4-high and CHRNB4-low patients, respectively; *c* and *d* are the numbers of the patients with a wild-type gene in CHRNB4-high and CHRNB4-low patients, respectively; *n* is the total number (i.e., *a* + *b* + *c* + *d*) of patients. For example, there were 25 and 8 patients with altered PIK3CA in CHRNB4-high and CHRNB4-low smoking patients, thus the counts of *a* and *b* were 25 and 8, respectively; furthermore, the counts of *c* and *d* were 58 and 80 for the patients with wild-type PIK3CA in two subgroups. Next, the Fisher’s exact *p*-value of each gene was measured for identifying 246 significantly different distributions of CNA (when *p* < 0.05) between CHRNB4-high/-low groups (only 83 of 87 CHRNB4-high patient’s CNA were obtained from database, Appendix A).

### 4.7. Cell Lines

In this study, three HNSCC cell lines from the previous work, FaDu, SCC25 and OECM1, with and without NNK treatment [34], were used to explore the HNSCC mechanism underlying nicotine mediation. The FaDu and SCC25 cell lines were obtained from the American Type Culture Collection (ATCC numbers CRL-1628 and HTB-43, respectively), and OECM1 cell line was obtained by Dr. Lee from the Department of Otolaryngology-Head and Neck Surgery, Tri-Service General Hospital, National Defense Medical Center, Taipei, Taiwan. Three cell lines were cultured in Roswell Park Memorial Institute (RPMI) medium supplemented with 10% foetal bovine serum at 37 °C in the presence of 5% CO_2_. 4-Methylnitrosamino-1-3-pyridyl-1-butanone (NNK) was acquired from ChemSyn Laboratories (Lenexa, KS) and used to treat FaDu, SCC25, and OECM1 parental cells at a concentration of 500 nM for at least 1 month.

### 4.8. CRISPR/Cas9 Gene Editing on CHRNB4 and Lentivirus Production

To verify the role of CHRNB4 in smoking patients, we performed CRISPR/Cas9 gene editing in NNK long-term treated FaDu and SCC25 cell lines. The sgRNA of CHRNB4 was designed by using the MIT CRISPR Design website (http://crispr.mit.edu). According to the Zhang laboratory protocol [93], guide oligonucleotides were phosphorylated, annealed, and cloned into the BsmBI site of the lentiCRISPR v2 vector (Addgene #52961, kindly provided by F. Zhang lab). The CHRNB4 sgRNA-1, CHRNB4 sgRNA-2, and scramble-sgRNA contained plasmids that were verified by Sanger sequencing. The plasmids were cotransfected with pMD2.G (Addgene plasmid #12259) and psPAX2 (Addgene plasmid #12260), which were both kindly provided by Didier Trono, EPFL, Lausanne, Switzerland. Lentiviral particles were collected at 36 and 72 h and then concentrated with a Lenti-X Concentrator (Clontech, Mountain View, CA, USA) [86,94,95,96]. The lentiviral particles were analysed by Q-PCR [30]. A total of 1 × 10^6^ HNSCC cells were plated in a 6 cm dish and treated with lentivirus at an MOI of 5. Two days after transfection, the medium was replaced with medium containing 2.5 μg/mL puromycin for 2 days. Lentivirus-transfected cells were recovered 2 days before the experiments.

### 4.9. Western Blot Analysis

We used 12% sodium dodecyl sulfate polyacrylamide gel electrophoresis to separate whole-cell lysates by electrophoresis, which were then transferred to polyvinylidene fluoride membranes. The membranes were blocked with 5% non-fat milk at room temperature for 1 h. The primary antibody of CHRNB4 (Proteintech, 22192-1-AP) was used to probe the membranes. We prepared Tris-buffered saline-Tween-20 buffer with 3% non-fat milk to incubate the membrane at 4 °C overnight, and peroxidase-conjugated rabbit anti-goat secondary antibody was added (Santa Cruz Biotechnology, Dallas, TX, USA) at room temperature for 1 h. The immunoblots and luminescence were visualized by an enhanced chemiluminescence system and X-ray film.

### 4.10. Migration and Invasion Assays

Transwell migration and invasion assays were performed as follows. Parental and NNK-treated HNSCC cells were suspend in culture medium containing 0.5% serum and then plated in the upper chamber. For the invasion assay, the Matrigel–medium (1:2) mixture was supplied onto the membrane of the upper chamber before the cells were seeded. After 24 h incubation for the migration assay and 48 h incubation for the invasion assay, cells on the upper side of the filter were removed, and migrated and invaded cells were fixed in methanol and stained with haematoxylin. The number of migrated/invaded cells were counted using light microscope.

## 5. Conclusions

We suggest that CHRNB4 is a prognostic indicator for smoking HNSCC patients on the basis of the finding that high CHRNB4 expression was associated with adverse outcomes. The genes involved in angiogenesis, proliferation, and cell survival were more frequently dysregulated in smoking patients with higher CHRNB4 expression than in other patient groups. In addition, we confirmed the association between CHRNB4 and cell migration and invasion by using gene editing. Then, on the basis of drug repurposing approaches, we discovered that the smoking-cessation drug varenicline was able to reduce the migration and invasion of NNK-treated cells. Our methods and knowledge regarding CHRNB4 and its targeting agent could be useful for the development of precision medicine for smoking-related HNSCC.

## Figures and Tables

**Figure 1 cancers-12-01324-f001:**
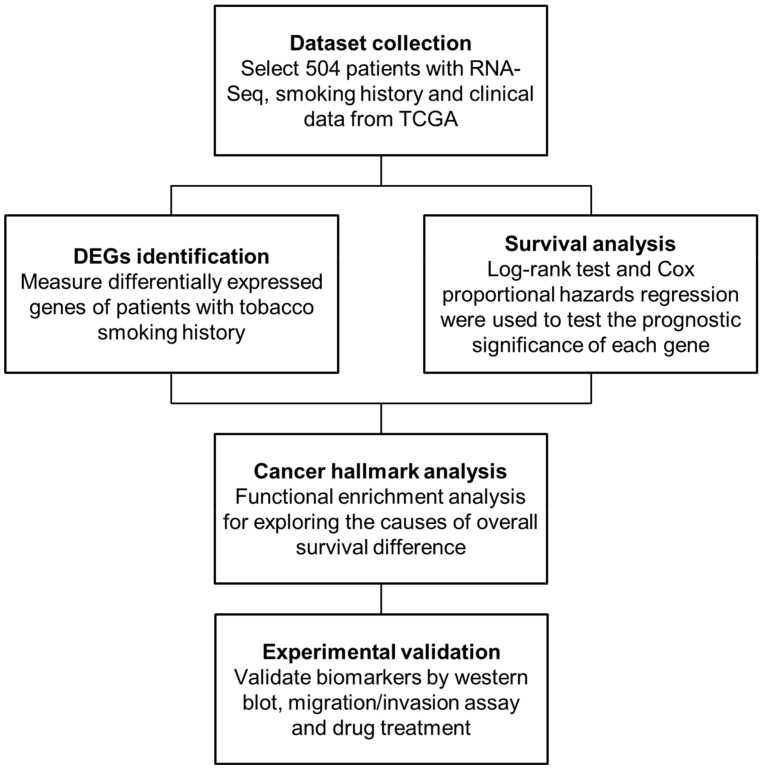
Main flowchart for identifying smoking-related biomarkers and repurposing drugs in head and neck squamous cell carcinoma (HNSCC). In total, 504 HNSCC patients, whose data included mRNA expression (RNA-Seq) data and clinical data (e.g., smoking history, survival state, and follow-up time), were selected from the The Cancer Genome Atlas (TCGA). Then, the differentially expressed genes were identified and survival analysis was investigated to determine the clinical outcome of smoking patients. Third, the enrichment analysis of cancer hallmarks was examined to understand the causes of poor prognosis in smoking patients. Finally, two HNSCC cell lines were utilized to validate gene candidates and drugs for their therapeutic potential.

**Figure 2 cancers-12-01324-f002:**
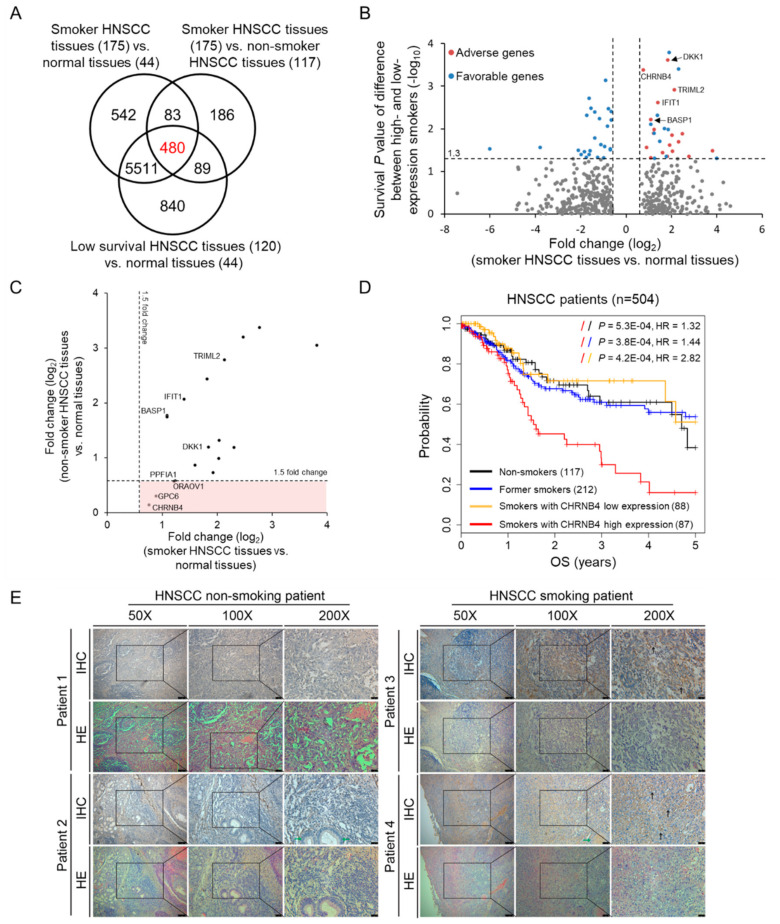
The strategies for determining HNSCC smoking-related and prognostic biomarkers. (**A**) Venn diagram of differentially expressed genes (DEGs) among normal samples, and HNSCC samples from smoking, non-smoking, and low-survival patients. (**B**) Identification of 18 adverse (red) and 36 favourable (blue) prognostic genes from 480 DEGs by survival analysis based on high- and low-expression of each gene. (**C**) Fold change of 18 adverse genes comparing smoker (*x*-axis) and non-smoker (*y*-axis) HNSCC tumours to normal tissues. Four genes are located in red region because they are differentially expressed between smoker tumours and normal tissues, but are not between non-smoker tumours and normal tissues. (**D**) Kaplan–Meier plots of overall survival (OS) of 504 HNSCC patients, including non-smokers (black), former smokers (blue), smokers with low neuronal acetylcholine receptor subunit beta-4 (CHRNB4) expression (orange), and smokers with high CHRNB4 expression (red). (**E**) Immunohistochemistry stain of clinical HNSCC patients. The tumour region tissue slides from smoking and non-smoking HNSCC patients were stained with CHRNB4 primary antibody, whereas all adjacent sections and same slides were counterstained with haematoxylin and eosin (HE stain) for general histological orientation. Intensive membrane CHRNB4 expression on cancerous region is black arrowed, whereas the CHRNB4 expression in normal adjacent region is indicated by a green arrow.

**Figure 3 cancers-12-01324-f003:**
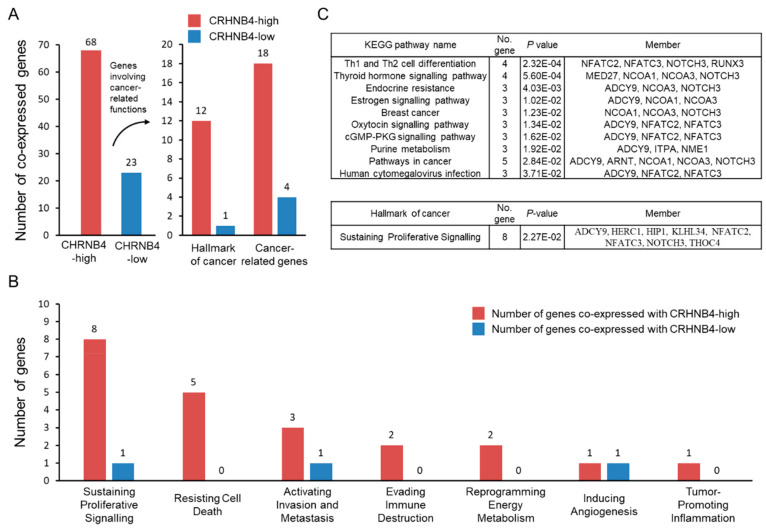
Association between cancer-related functions and CHRNB4 co-expressed genes of CHRNB4-high/low patients. (**A**) Number of co-expressed genes with CHRNB4 in the CHRNB4-high and -low subgroups involved in cancer hallmark and cancer-related functions. (**B**) The histogram of co-expressed genes involved in seven cancer hallmarks in CHRNB4-high (red) and CHRNB4-low (blue) smoking patients. (**C**) Kyoto Encyclopedia of Genes and Genomes (KEGG) pathway and cancer hallmarks enrichment analysis of co-expressed genes in the CHRNB4-high subgroup.

**Figure 4 cancers-12-01324-f004:**
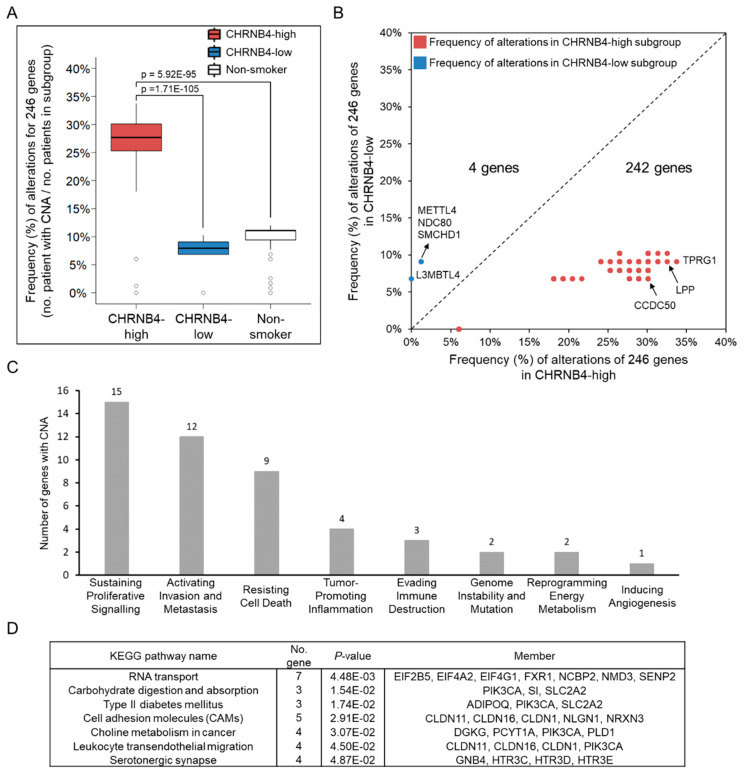
Copy number alterations (CNAs) between the CHRNB4-high and -low subgroups. (**A**) The boxplots of 246 differential CNAs in CHRNB4-high smoking (red), CHRNB4-low smoking (blue), and non-smoking (white) patients. The two-sided *p*-value was calculated with Student’s *t*-test. (**B**) Scatter plots of CNAs between the CHRNB4-high (*x*-axis) and CHRNB4-low (*y*-axis) smoking patients. The genes below the diagonal line indicate that the genes had a higher frequency of genetic alterations in the CHRNB4-high subgroup (red) than in the CHRNB4-low subgroup (blue). (**C**) The histogram of eight cancer hallmarks of genes with differently frequent CNA between CHRNB4-high and CHRNB4-low subgroups. (**D**) KEGG pathway enrichment analysis of 246 copy number-altered genes.

**Figure 5 cancers-12-01324-f005:**
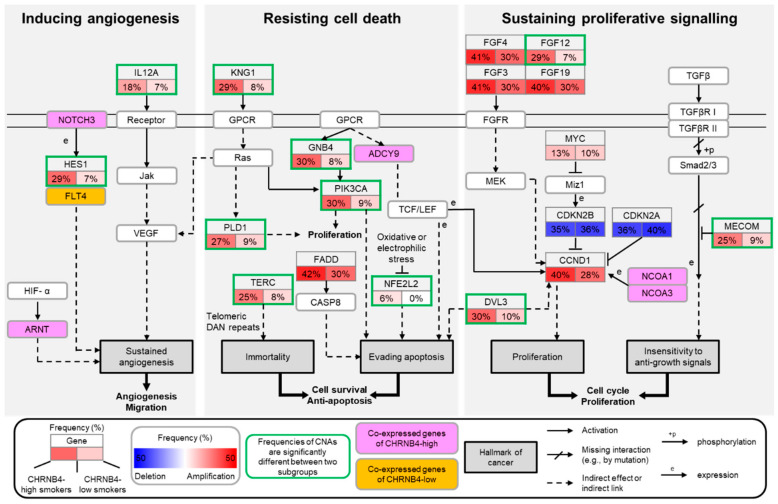
Graphic illustration of the identified genes, functions, and pathways of tumorigenesis in CHRNB4-high and CHRNB4-low subgroups. The key genes, limited number of pathways, and inferred functions for tumorigenesis are based on CHRNB4-high and CHRNB4-low smokers. Subpanels indicate the frequency (%) of copy number alteration (CNA) in CHRNB4-high (left subpanel) and CHRNB4-low subgroups (right subpanel). In each subgroup, the subpanels display the difference (subtraction) of the frequencies between amplification and deletion, and are coloured red if the frequency of amplification is greater than the frequency of deletion, otherwise they are coloured blue. The green border indicates a gene whose CNA frequency between CHRNB4-high and -low subgroups was significantly different (examined by Fisher’s exact test, *p* < 0.05). Five genes (e.g., NOTCH3, ARNT) are coloured pink, and one gene (FLT4) is orange, representing the genes co-expressed with CHRNB4 in CHRNB4-high and -low subgroups, respectively.

**Figure 6 cancers-12-01324-f006:**
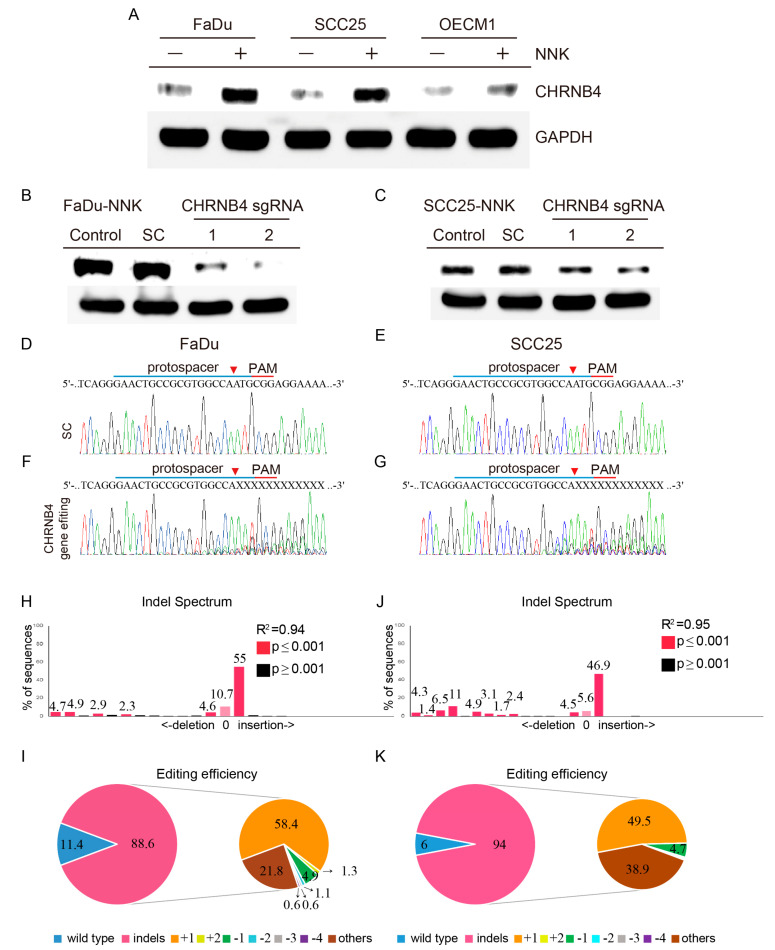
Validation of CHRNB4 gene editing in FaDu and SCC25 HNSCC cells using the CRISPR/Cas9 system. (**A**) The induction of CHRNB4 protein expressions on long-term nicotine-derived nitrosamine ketone (NNK)-treated HNSCC cells. The FaDu, SCC25, and OECM1 cell lines were treated NNK for 1 month, followed by performing Western blot to compare the CHRNB4 protein expression with their parental controls. The protein validations of CHRNB4 gene editing were performed by CHRNB4 sgRNA-1, sgRNA-2, and scramble-sgRNA virus transfected into long-term NNK-treated (**B**) FaDu and (**C**) SCC25 cells. Schematic representation of the human CHRNB4 protospacer sequence (blue underline) for gene editing. The arrowhead indicates the expected Cas9 cleavage site. The protospacer adjacent motif (PAM, red underline) is the motif required for Cas9 nuclease activity. CHRNB4 sgRNA-2 and scrambled (SC) sgRNA were delivered to FaDu (**D**,**F**) and SCC25 (**E**,**G**) cells by lentivirus. After transduction, DNA from virus-infected cells was purified and subjected to Sanger sequencing of CHRNB4 gene locus. The Tracking of Indels by Decomposition (TIDE) analysis is shown for CHRNB4 sgRNA-2 virus transfected in FaDu (**H**) and SCC25 (**J**) cells, compared to SC control. The pie charts show the percentages of indels in the CHRNB4 gene editing by CHRNB4 sgRNA-2 lentivirus on FaDu (**I**) and SCC25 (**K**) cells. The gene editing efficiency of the two cell lines are presented in pink, while the two most common +1 indels and other mutations are presented in orange and brown, respectively.

**Figure 7 cancers-12-01324-f007:**
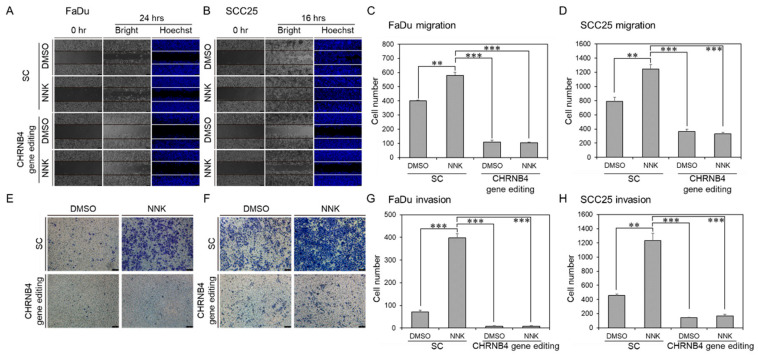
Migration and invasion effects in parental and NNK-treated HNSCC cell lines with CHRNB4 gene editing. The migration and invasion assays of long-term NNK-treated (**A**,**E**) FaDu and (**B**,**F**) SCC25 cell lines with or without 0.1 μM NNK treatments were conducted and recorded. The migration and invasion of (**C**,**G**) FaDu and (**D**,**H**) SCC25 cell lines were counted and presented in a histogram. The experiments were repeated three times, and a two-sided *p*-value was calculated with Student’s *t*-test. The error bars indicate the standard error. *p*-values less than 0.01, and 0.001 are denoted by **, and ***, respectively.

**Figure 8 cancers-12-01324-f008:**
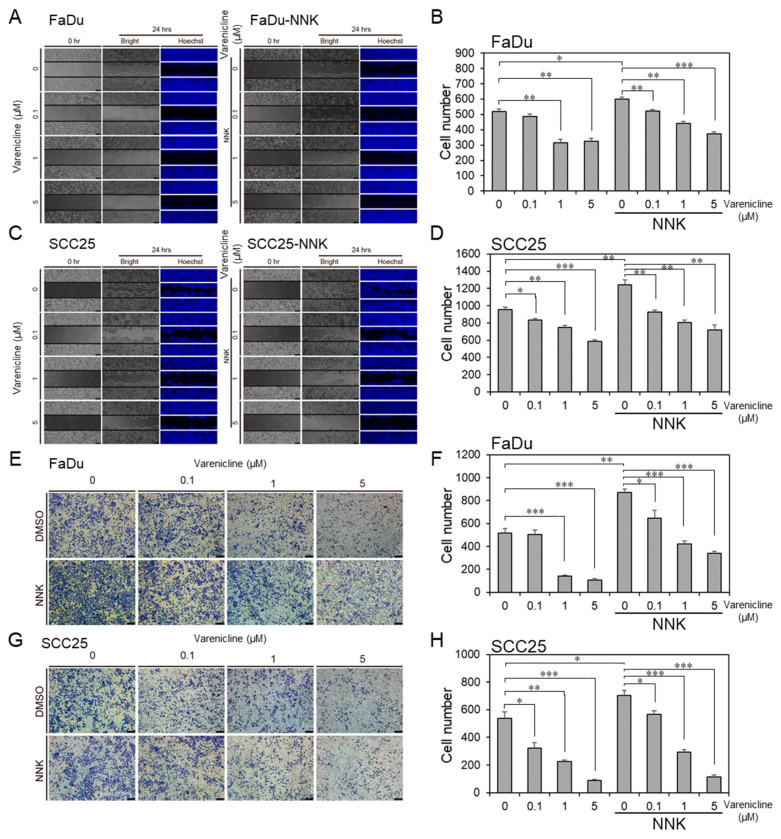
An FDA-approved drug, varenicline, inhibits migration and invasion in NNK-treated cells. Migration ability was measured after 24 h of varenicline treatment at 0.1, 1, and 5 µM in (**A**,**B**) FaDu and (**C**,**D**) SCC25 cell lines with or without NNK treatments. Invasion ability was measured after 48 h of varenicline treatment at 0.1, 1, and 5 µM in (**E**,**F**) FaDu and (**G**,**H**) SCC25 cell lines with or without NNK treatments. The experiments were repeated three times, and a two-sided *p*-value was calculated with Student’s *t*-test. The error bars indicate the standard error. *p*-values less than 0.05, 0.01, and 0.001 are denoted by *, **, and ***, respectively.

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
