# Peer review of "An Integrated Genomic Strategy to Identify CHRNB4 as a Diagnostic/Prognostic Biomarker for Targeted Therapy in Head and Neck Cancer"

_cancers, 2020, doi:10.3390/cancers12051324_

Round 1

Reviewer 1 Report

The authors have addressed all of my critiques raised during the previous round of review. However, the manuscript would have to undergo language editing for clarity (several sentences are grammatically incorrect). No other concerns noted. I would recommend acceptance.

Reviewer 2 Report

I went through the rebuttal document, and yes, the authors have appropriately addressed my original concerns with the additional experiments.

This manuscript is a resubmission of an earlier submission. The following is a list of the peer review reports and author responses from that submission.

Round 1

Reviewer 1 Report

The manuscript by Chuang et al., reports on the potential of CHRNB4 as a diagnostic and/or prognostic biomarker in smoking associated HNSCC patients.  The overall significance of the work is high. The experiments and bioinformatics analysis are well designed and presented. However, several issues in description of the methodology and results warrant further attention by the authors.

1. Introduction, 1st paragraph: The references 1-4 are not direct studies on epidemiology of HNSCC. Please use appropriate reference (e.g. GLOBOCAN report).

2. Introduction, 2nd paragraph: The authors provide an overview of smoking associated HNSCC. However, several statements need to be deleted or rephrased. For example, “To reveal the role of smoking in HNSC is an urgent task’. The role of smoking in HNSCC (e.g. formation of DNA adducts) biology, therapeutic response) has been well established and extensively studied.

3. Introduction, 3rd paragraph: The description provided by the authors re: nicotine and nAChRs is relevant. What is the current clinical evidence on the nACHRs in HNSCC? Is there an association between nACHR and disease progression or therapeutic response? Please review and summarize findings from current literature: Carracedo et al. 2007 PMID: 17465209; Scherl et al. 2016; PMID: 27234470.

4. Introduction, Paragraphs 4,5: The statements regarding the need for developing response biomarkers/signatures needs to be rephrased (ref. 40 is not directly relevant to the stated claims). There is a considerable body of work by several groups that have reported potentially useful signatures of therapeutic resistance/sensitivity in HNSCC that need to be recognized. Statement regarding lack of pharmaceutical targets for HNSCC is incorrect.

5. Figure 2. The authors state that Cox proportional hazards analysis was employed to identify the prognostic genes. What were the confounding variables and how were they adjusted in the analysis. Some details are required.

6. Figure 2C. As presented, the 4-gene signature (CHRNB4, GP6, ORAOV1, PPFIA1) appears to downregulated in both smokers and non-smokers when compared to normal tissues. Some clarification is needed.

7. In vitro experiments (Fig 6 and Fig. S2): The western blot (Fig. S2) of untreated FaDu and SCC25 cells do not show detectable expression of CHRNB4 in HNSCC cells. It is therefore unclear how the authors confirmed the successful k/d or k/o in these cell lines. The authors should include non-template controls for both siRNA and sgRNA treatment since the transfection could confound results. Also, include the size of CHRNB4 as it is unclear from the presented western blot.

8. The impact of siRNA and sgRNA against CHRNB4 was also seen only in the cells following treatment with NNK not the untreated controls. Statements regarding the effect of varenicline in the parental vs. NNK treated cells should be clearly explained and the results discussed.

9. The authors could also consider include two HPV+ HNSCC cell lines to strengthen their findings. An vivo study to demonstrate the potential of Varenicline would also be useful. '

10. CHRNB4 in human HNSCC: Although analysis of the RNA seq data from the TCGA HNSCC dataset supports the potential utility of CHRNB4 as a prognostic biomarker, no evidence regarding protein expression of CHRNB4 (using immunohistochemistry) in human HNSCC is presented. This would strengthen their findings. Alternatively, the authors should recognize the limitations of using existing datasets.

Additional comments:

Change ‘HNSC’ to ‘HNSCC’.

Several references cited by the authors are not directly pertinent to the statements or are incorrect.

Reviewer 2 Report

The authors present an interesting story discovering a role for CHRNB4 in HNSCC (called HNSC in the paper), that highlights its association with low survival and smoking status, performs GSEA type analysis on the correlated CNVs and genes and then performs functional experiments to test the role of the gene in motility/invasion in two cell line models. Finally, the authors claim to identify a drug that targets the pathway and phenocopies CHRNB4 siRNA/sgRNA. 

Overall, the paper proposes many interesting discoveries, but lacks sufficient depth in each figure to determine if the claims are accurate. The analysis as completed is highly superficial and lacks rigor. Some major comments are attached to help the authors improve the manuscript, but substantial revisions and additional controls would be required to re-consider this manuscript. 

1) The authors should validate the relationship of expression status in tissues by IHC, or if antibodies are not available, then by RNAscope. If this is not done, then the meaning of subsequent experiments is unclear. 

2) The authors should validate knockdown/knockout efficiency in the models used for functional assays. Do the constructs/siRNAs also inhibit migration and invasion in models with low CHRNB4 expression (e.g. a negative control?). Does overexpression drive motility and invasion?

3) Does RNAseq or qPCR analysis of gene sets from the  knockdown/knockout experiment validate the CHRNB4-related signature? 

4) The pharmacogenetic link between the role of CHRNB4 and varinicline in these models should be demonstrated more rigorously. 

5) what is the mechanism by which CHRNB4 regulates motility or invasion?